# Estimating the Potential of Radiomics Features and Radiomics Signature from Pretherapeutic PSMA-PET-CT Scans and Clinical Data for Prediction of Overall Survival When Treated with ^177^Lu-PSMA

**DOI:** 10.3390/diagnostics11020186

**Published:** 2021-01-28

**Authors:** Sobhan Moazemi, Annette Erle, Susanne Lütje, Florian C. Gaertner, Markus Essler, Ralph A. Bundschuh

**Affiliations:** 1Department of Nuclear Medicine, University Hospital Bonn, 53127 Bonn, Germany; annette.erle@t-online.de (A.E.); susanne.luetje@ukbonn.de (S.L.); florian.gaertner@ukbonn.de (F.C.G.); markus.essler@ukbonn.de (M.E.); ralph.bundschuh@ukbonn.de (R.A.B.); 2Department of Computer Science, University of Bonn, 53115 Bonn, Germany

**Keywords:** prostate cancer (PC), prostate specific membrane antigen (PSMA), positron emission tomography (PET), computed tomography (CT), overall survival (OS), radiomics signature (RS)

## Abstract

Prostate-specific membrane antigen (PSMA) positron emission tomography/computed tomography (PSMA-PET/CT) scans can facilitate diagnosis and treatment of prostate disease. Radiomics signature (RS) is widely used for the analysis of overall survival (OS) in cancer diseases. This study aims at investigating the role of radiomics features (RFs) and RS from pretherapeutic gallium-68 (^68^Ga)-PSMA-PET/CT findings and patient-specific clinical parameters to analyze overall survival of prostate cancer (PC) patients when treated with lutethium-177 (^177^Lu)-PSMA. A cohort of 83 patients with advanced PC was retrospectively analyzed. Average values of 73 RFs of 2070 malignant hotspots as well as 22 clinical parameters were analyzed for each patient. From the Cox proportional hazard model, the least absolute shrinkage and selection operator (LASSO) regularization method is used to select most relevant features (standardized uptake value (SUV)_Min_ and kurtosis with the coefficients of 0.984 and −0.118, respectively) and to calculate the RS from the RFs. Kaplan–Meier (KM) estimator was used to analyze the potential of RFs and conventional clinical parameters, such as metabolic tumor volume (MTV) and standardized uptake value (SUV) for the prediction of survival. As a result, SUV_Min_, kurtosis, the calculated RS, SUV_Mean_, as well as Hemoglobin (Hb)1, C-reactive protein (CRP)1, and ECOG1 (clinical parameters) achieved *p*-values less than 0.05, which suggest the potential of findings from ^68^Ga-PSMA-PET/CT scans as well as patient-specific clinical parameters for the prediction of OS for patients with advanced PC treated with ^177^Lu-PSMA therapy.

## 1. Introduction

Computer-aided diagnosis (CAD), leveraging state-of-the-art statistical methods, has gained critical importance in diagnosis and therapy planning in recent years. Radiomics features (RFs) extracted from base-line prostate-specific membrane antigen (PSMA) positron emission tomography/computed tomography (PSMA-PET/CT) scans, together with patient-specific clinical parameters, can facilitate diagnosis and treatment of prostate disease [1,2]. As PSMA-PET/CT is gaining importance in the diagnosis and treatment planning of patients with advanced prostate cancer [3], obtaining additional information using radiomics features is of great interest, especially in the use of therapy planning [4]. Within the concept of theranostics, using the same ligand to PSMA either labeled with the positron emitter gallium-68 (^68^Ga) or the therapeutic beta-emitter lutethium-177 (^177^Lu) [5], this question is of special interest as one may speculate that the distribution of the ^68^Ga-PSMA may predict therapeutic use of ^177^Lu-PSMA in a symbiotic way.

In this study, we investigate the role of conventional parameters and RS from pretherapeutic ^68^Ga-PSMA-PET/CT findings, as well as patient-specific clinical parameters, to analyze overall survival (OS) of PC patients when treated with ^177^Lu-PSMA. First, the study cohort will be introduced. Then the study set-up, including the third-party as well as in-house developed software, will be described in detail. Finally, based on the results, the potential of each feature or parameter to predict the OS will be analyzed and quantified. To the best of our knowledge, the presented findings would be novel for the analysis of survival for PSMA-PET/CT, although for other modalities such as MRI or other markers such as FDG-PET, there is quite some work [6,7,8,9].

## 2. Materials and Methods

### 2.1. Methodological Background

Conventionally, variables such as metabolic tumor volume (MTV), total lesion Glycolysis (TLG), and standardized uptake value (SUV)_Mean/Max_ have been in focus for the analysis of overall survival (OS). For example, analysis of Fluorodeoxyglucose (FDG)-PET/CT for non-small-cell lung cancer (NSCLC) [10], FDG-PET for esophageal cancer patients [11], and FDG-PET/CT for patients with relapsing/refractory non-Hodgkin lymphoma [12]. Moon et al. [11] also provided a summary of literature for PET or PET/CT analyses based on TLG, MTV, and SUV_Mean/Max_.

For the analyses of single features, Kaplan–Meier (KM) statistics [13] are applied which subdivide the study cohorts based on cut-off values of the features. The cut-off values are either chosen based on previous findings or by subdividing the cohorts into subgroups with high- and low- risk (and sometimes middle). Consecutively, the *p*-values of the KM statistics are calculated based on log-rank tests [14] to find significantly correlating features with overall survival.

As the processing power to apply more sophisticated statistical methods on bigger data-sets advanced, simultaneous analyses of more variables became more popular. As a result, there is a wide set of computer-based tools and software packages [15,16] to support physicians and scientists to conduct time-to-event analyses more accurately. Additionally, for the prediction of OS, radiomics features (RFs) extracted from different imaging modalities can be processed by the software packages to quantify their potential for the OS analysis. For example, the Cox proportional hazard model (CPH) [17] is used in many studies for this purpose [6,7,8,9].

Dealing with the problem of having fewer subjects than features in the field of radiomics, the so-called term “radiomics signature” (RS) is referred and used in a bunch of studies for MRI [6,7,8] and FDG-PET/CT for lung cancer [9]. Most of the studies in this group deal with large numbers (often more than 1000) of radiomics features. Here, as a common approach, first a stepwise feature selection is applied followed by least absolute shrinkage and selection operator (LASSO) [18] to end up with the radiomics signature. Then the survival prediction metrics (like KM or cumulative hazard) are used to assess and quantify the predictive outcome of the calculated RS and compare it with that of the conventional or the clinical parameters.

### 2.2. Patients and Volume of Interest (VoI) Definition and Annotation

83 male patients who had been histologically diagnosed with prostate carcinoma and were referred for ^68^Ga-PSMA PET/CT were included in this retrospective study. An intravenous injection of 98 to 159 MBq in-house produced ^68^GA-HBED-CC PSMA carried out about 40 to 80 minutes before each base-line scan. To acquire the low-dose CT (16mAs, 130 kV) from the base of skull to mid-thigh, as well as the PET scan over the same area with 3 or 4 minutes per bed position based on the body weight of the patient, a Biograph 2 PET/CT system (Siemens Medical Solutions, Erlangen, Germany) was used. The PET and CT data were reconstructed in 128 × 128 and 512 × 512 matrices, respectively. Both PET and CT data had 5 mm slice thicknesses. An attenuation-weighted ordered subsets expectation maximization algorithm was used for attenuation and scatter corrections (8 iterations, 16 subsets) and a 5 mm Gaussian post-reconstruction-filter was applied afterwards by the manufacturer.

A trained nuclear medicine (NM) physician (board certified with 7 years’ experience in PET/CT analysis) identified and delineated all the pathological hotspots for each scan, using InterView Fusion software (Mediso Medical Imaging, Budapest, Hungary, Version 3.08.005) which resulted in 2070 pathological hotspots. All the primary tumors if present as well as all the metastatic uptakes in all of the organs were identified as the hotspots. A total of 73 (37 PET-based + 36 CT-based) radiomics features, including first and higher order statistics (mean, max, kurtosis, etc.), shape-based (max diameter), textural (entropy, contrast, homogeneity, etc.), and volumetric zone and run length statistics features (grey-level non-uniformity, short run emphasis, etc.) were calculated for each hotspot (Table 1).

Furthermore, for each individual patient, 14 numerical (such as age, weight, and height) as well as 8 categorical therapeutic clinical parameters (such as Gleason score, ECOG1, and ALP1) were included (Table 2).

All the numerical variables were standardized prior to analyses steps using the MinMaxScaler method provided by SciKit-Learn library [19].

For the imaging procedure and for the anonymized evaluation of the data, all the patients gave written and informed consent. Due to the retrospective character of the data analysis, an ethical statement was waived by the institutional ethical review board.

### 2.3. Statistical Analyses

For the statistical analyses pipeline, both of the univariate and multivariate approaches for the analysis of overall survival were considered. First, the Cox proportional Hazard model was used as the multivariate method [20] to analyze the radiomics features and also to select the most significant features, which results in the calculation of the radiomics signature using the LASSO method. Consecutively, linear regression tests were performed to confirm the significance of the selected variables and that of RS. This was followed by KM statistics to assess the predictive outcome of the variables for OS analysis.

To form the standard structured input for the survival analysis pipeline with right-censoring, the information about the time of death for the patients who had died by the date on which the study began, or if the patients were still alive on that date, were taken into account. The standard structured survival information included two parameters: One Boolean variable indicating the status of the patient on the date the experiment started (dead = True or alive = False) and one integer variable indicating the number of months the patient had lived until the time of death or censoring, respectively.

As the number of input variables of the dataset (a total of 95 variables including 73 radiomics features, 14 numerical, and 8 categorical clinical parameters) exceeded the number of subjects (83), it was reasonable to apply feature selection prior to the survival analysis. To this end, from the Cox proportional hazard model provided by glmnet library of R programming [12], the least absolute shrinkage and selection operator (LASSO) method, also known as L1 regularization [18], was applied. This method was used to identify the most relevant features for the prediction of overall survival (OS). Furthermore, it provided coefficients for the selected variables, which were consecutively used to calculate the so-called radiomics signature (RS) for each subject.

To confirm the significance of the selected RFs by LASSO method, as well as that of the calculated RS, linear regression was applied. To this end, the normalized values of the selected variables and the RS were analyzed in connection with the overall survival in months.

To achieve more interpretable outcomes, from the survival analyses package from SciKitSurvival, the Kaplan–Meier (KM) estimator [21] was used. Based on predefined cut-off values, the KM estimator helps to analyze whether different groups of subjects separated by different values of a given variable have significantly different survival times. To avoid ending up with too small groups of subjects which would affect the generalizability of the results, the median value of the numerical variables and different possible values of the categorical variables were used as the cut-off values for this study.

As the final step, the predictive performance of the conventional features (e.g., MTV and SUV_Mean/Max_) as well as the clinical parameters are compared to that of the selected variables and the calculated RS as provided by the LASSO method. An overview of the study pipeline is shown in Figure 1.

## 3. Results

### 3.1. Clinical Characteristics of the Patient Cohort

The age range of the study cohort varied between 48 and 87 years and the serum PSA levels ranged from 4.7 to 5910 ng/mL. Their Gleason scores also varied between 6 and 10. The baseline ^68^Ga-PSMA PET/CT scans were carried out from November 2014 to August 2019 and their ^177^Lu-PSMA treatment followed five to 21 days thereafter. Table 3 summarizes the clinical characteristics of the participants and gives an overview about the therapeutic aspects of the study cohort.

### 3.2. Selected Features and Radiomics Signature

From the 73 radiomics features, the LASSO method identified SUV_Min_ and kurtosis as the most correlating features with the overall survival time. The corresponding coefficients of 0.984 and −0.118 were calculated, respectively. To form the radiomics signature of each patient the calculated coefficients were used as follows:(1)$$RS_{\left(i\right)}=SUV_{Min\;\left(i\right)}\times 0.984\;+Kurtosis_{\left(i\right)}\;\times \left(-0.118\right)$$
where the $RS_{\left(i\right)}$ is the radiomics signature for the patient number i and $SUV_{Min\;\left(i\right)}\;$ and $Kurtosis_{\left(i\right)}\;$ are the values of the selected variables for the patient number i
i (Equation (1)).

Regression tests results confirm the significance of the selected RFs and the calculated RS, as they achieve *p*-values less than 0.05. Figure 2 illustrates the regression diagrams for the selected variables as well as the calculated RS.

### 3.3. Survival Prediction

Among all the variables included in the Kaplan–Meier estimation experiment, SUV_Min_, kurtosis, the calculated RS, SUV_Mean_, as well as three clinical parameters (Hb1, CRP1, and ECOG1) achieved *p*-values less than 0.05. Figure 3 shows the Kaplan–Meier diagrams of these variables. 

## 4. Discussion

Treatment of advanced prostate carcinoma using ^177^Lu-PSMA is gaining importance and numbers of treatments are increasing recently. However, not all patients are responding well and about one third of treatments fail [22]. Therefore, the aim of this study was to analyze radiomics and clinical features for their predictive value for OS of prostate cancer patients undergoing ^177^Lu-PSMA treatment.

Out of 73 radiomics features, we identified the most relevant ones for the analysis of OS by means of the state-of-the-art statistical methods. First, the multivariate feature selection method (LASSO) identified SUV_Min_ and kurtosis as the most important variables among all the 73 radiomics features, which together formed the radiomics signature. Then, Kaplan–Meier statistics was applied to quantify the predictive potential of the radiomics signature as well as each individual radiomics feature or clinical parameter. Our findings confirm the important role of the conventional parameters, such as SUV. This may be surprising, as we found in other studies that conventional parameters, such as mean SUV or maximum SUV, did not show predictive power [22,23]. Another surprising point is that the minimum SUV seems to have predicting power. The reason for this may be that the low SUV values within the segmented volume may represent also some kind of heterogeneity, as it may correlate with a broader spectrum of activity values within a tumor. In addition, textural heterogeneity parameters, such as kurtosis, taken from pretherapeutic ^68^Ga-PSMA-PET/CT scans were found as strong parameters in this study, which is in accordance with previous findings [4]. However, in this study the parameters were just correlated individually. 

Although the study outcome reveals the potential of radiomics signature and textual heterogeneity parameters, even in the absence of clinical parameters, the importance of the clinical parameters such as Hb and especially ECOG is shown. This corresponds to the study by Ferdinandus et al. who also found, for example, Hb to be a predictive parameter [22]. These findings suggest that, as much as possible, clinical information needs to be included in therapy decision support systems and not only obvious data, such as imaging data or tumor markers.

We included 2070 pathological hotspots from 83 subjects as well as a total of 95 variables in this study. The results illustrate the potential of the investigated variables and the corresponding statistical methods to address the problem of overall survival prediction for patients with advanced prostate carcinoma. However, to make the results more generalizable, experiments with larger cohorts need to be conducted.

The annotations of the ^68^Ga-PSMA-PET/CT findings by an experienced NM physician are used as the ground truth instead of histopathological information. This might be considered as a drawback of the current study; however, considering the ethical perspectives regarding patient examinations, the non-invasive approach was chosen.

## 5. Conclusions

Radiomics features and radiomics signature from pretherapeutic ^68^Ga-PSMA-PET/CT scans as well as patient-specific clinical parameters hold promise for the prediction of overall survival for patients with advanced prostate carcinoma treated with ^177^Lu-PSMA therapy.

## Figures and Tables

**Figure 1 diagnostics-11-00186-f001:**
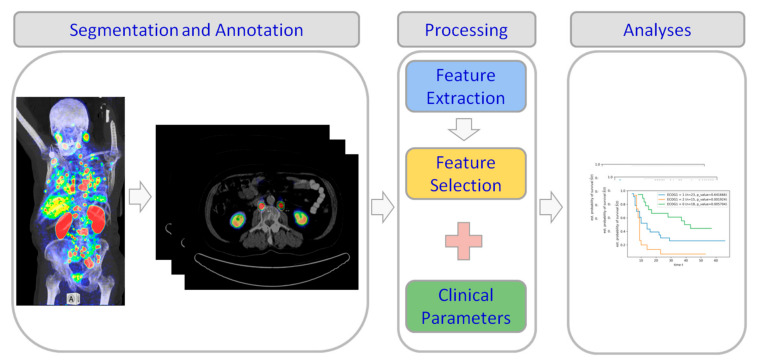
The study pipeline. First, the positron emission tomography/computed tomography (PET/CT) images are manually segmented and annotated by an experienced NM physician. Then the radiomics features are extracted and the most relevant features among them are chosen by LASSO method [18] to calculate the radiomics signature. Finally, the Kaplan–Meier estimator [20] is used to analyze and visualize the survival prediction results.

**Figure 2 diagnostics-11-00186-f002:**
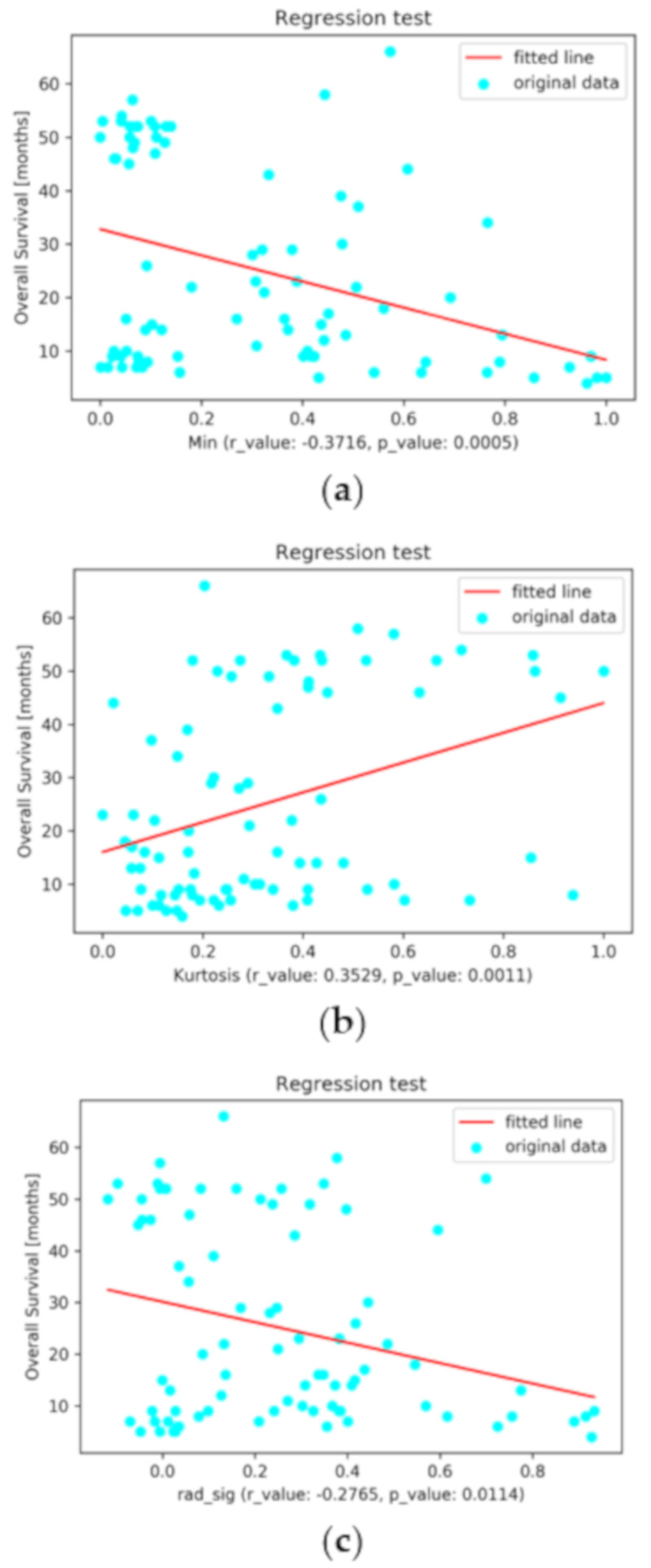
The linear regression diagrams for overall survival (in months) based on (**a**) SUV_Min_, (**b**) kurtosis, and (**c**) radiomics signature.

**Figure 3 diagnostics-11-00186-f003:**
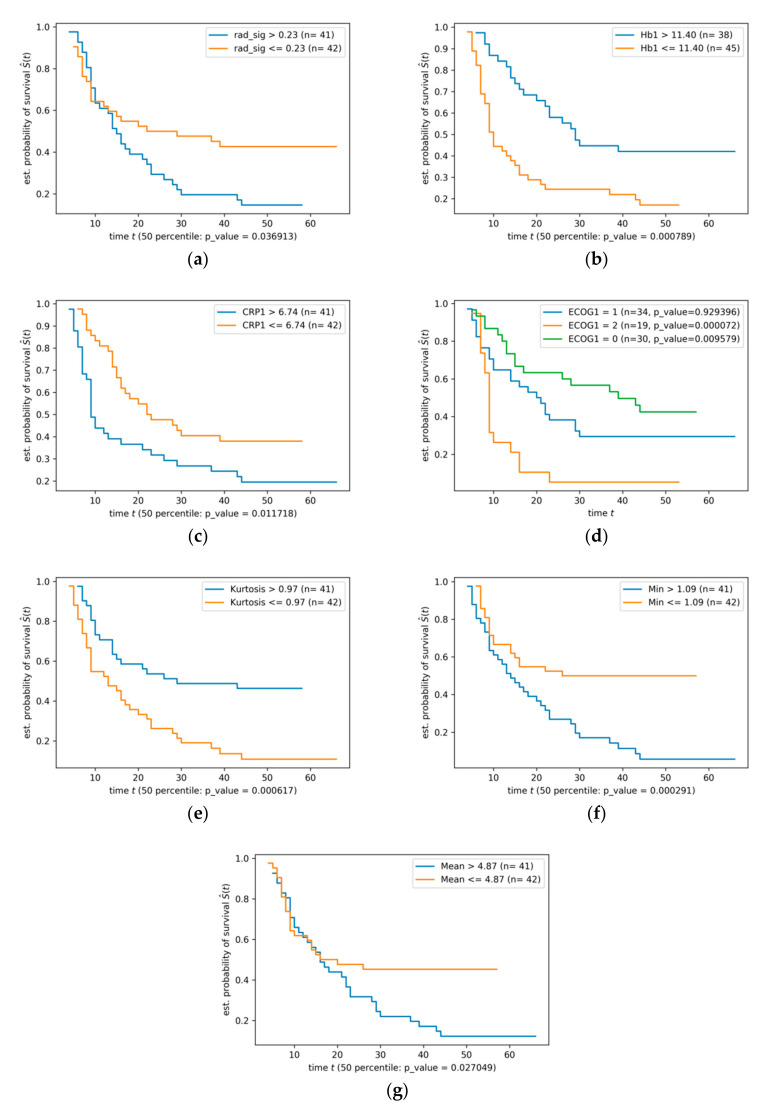
The results of Kaplan–Meier analyses for (**a**) radiomics signature, (**b**) Hb1, (**c**) CRP1, (**d**) ECOG1, (**e**) kurtosis, (**f**) SUV_Min_, and (**g**) SUV_Mean_.

**Table 1 diagnostics-11-00186-t001:** List of the radiomics features from both positron emission tomography (PET) and computed tomography (CT) modalities. Please note that the metabolic tumor volume (MTV) is PET-specific.

First or Higher Order Statistics	Shape and Size	Textural	Volumetric Zone Length Statistics	Volumetric Run Length Statistics
Deviation Mean Max Min Sum PET-MTV Kurtosis	Max. Diameter	Entropy Homogeneity Correlation Contrast Size Variation Intensity Variation Coarseness Busyness Complexity	Short Zone Emphasis Long Zone Emphasis Low Grey-Level Zone Emphasis High Grey-Level Zone Emphasis Short Zone Low Grey-Level Emphasis Short Zone High Grey-Level Emphasis Long Zone Low Grey-Level Emphasis Long Zone High Grey-Level Emphasis Zone Percentage	Short Run Emphasis Long Run Emphasis Low Grey-Level Run Emphasis High Grey-Level Run Emphasis Short Run Low Grey-Level Emphasis Short Run High Grey-Level Emphasis Long Run Low Grey-Level Emphasis Long Run High Grey-Level Emphasis Grey-Level Non-Uniformity Run Length Non-Uniformity Run Percentage

**Table 2 diagnostics-11-00186-t002:** Descriptions of the clinical parameters (PSMA: prostate specific membrane antigen, PET: positron emission tomography).

Parameter	Description
Age	Age at the first PSMA PET
Weight	Weight at the first PSMA PET
Height	Height at the first PSMA PET
Gleason Score	Describes abnormality degree of cancer cells in prostate
ALP1	Serum alkaline phosphatase at the first PSMA PET
PSA1	Serum PSA level at the first PSMA PET
Time Difference	Time between the first diagnosis and the first PSMA PET
Crea1	Serum creatinine at the first PSMA PET
GGT1	Gamma-glutamyltransferase at the first PSMA PET
CRP1	C-reactive protein in serum at the first PSMA PET
Hb1	Hemoglobin at the first PSMA PET
Erys1	Erythrocytes at the first PSMA PET
Thrombose1	Thrombocytes at the first PSMA PET
Leukos1	Leicocytes at the First PSMA PET
ECOG1	Scale of the performance status of the patient
Prostatectomy	whether the patient underwent prostatectomy
Hormonal therapy	whether the patient underwent hormonal therapy
Chemotherapy	whether the patient underwent chemotherapy
Bisphosphonate	whether the patient had taken bisphosphonates
Radiotherapy Prostate	whether the patient underwent radiotherapy of prostate
Radiotherapy Bones	whether the patient underwent radiotherapy of bones
Radiotherapy LN	whether the patient underwent radiotherapy of lymph nodes

**Table 3 diagnostics-11-00186-t003:** (**A**) Mean values and ranges of the clinical parameters and (**B**) the therapy information of the patients.

**(A) Clinical Characteristic**	**Mean Value**	**Range (Min.: Max.)**
Age [years] Weight [Kilograms] prostate specific antigen (PSA) level [ng/mL] Gleason Score Hb1 [g/dL] CRP1 [mg/L]	70 79 493 8 11 23.6	48:87 49:125 4.7:5910 6:10 6:13.6 0.2:275
**(B) Diagnosis or Therapy Type**	**Number of Patients**	
Prostatectomy Hormonal therapy Chemotherapy Bisphosphonates Radiotherapy of prostate Radiotherapy of bones Radiotherapy of lymph nodes	40 81 59 66 44 44 75	

## Data Availability

The data are not publicly available because, due to the German regulations regarding data protection, we cannot make data available online or send it. However, all data are available for revision on-site.

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
