# Peer review of "Estimating the Potential of Radiomics Features and Radiomics Signature from Pretherapeutic PSMA-PET-CT Scans and Clinical Data for Prediction of Overall Survival When Treated with 177Lu-PSMA"

_diagnostics, 2021, doi:10.3390/diagnostics11020186_

Round 1
Reviewer 1 Report
interesting manuscript on imaging and treatment of prostate cancer. no significant limitations.
Author Response
Thank you for your comments. We have revised the manuscript with regards to English language proficiency/Spell-check.
Reviewer 2 Report
In this study, the authors retrospectively enrolled 83 mCRPC patients submitted to PSMA PET/CT before 177Lu-PSMA therapy and analyzed the prognostic impact of radiomic features and clinical parameters in the prediction of OS.
The topic of the study is of high clinical relevance. To the best of my knowledge, no other studies using a radiomic approach are present in the literature on this kind of patient at the moment. However, I have some considerations to point out.
Major points:
1) My major concern is related to the absence of uni- and multivariable analyses in the statistical design. KM curves display the prediction of OS by a specific parameter once it is binarized. However, using the KM approach, the reader cannot understand the prediction of OS by the same parameter as a continuous variable. Similarly, without the use of multivariate analysis, the reader is not able to understand the eventual independence in the prediction of OS between the statistically significant parameters.
2) Results are scarcely detailed. In particular, a detailed description of the clinical characteristics of the study cohort is needed. A few clinical details are reported in the methods section. I suggest moving to results and implementing this section. A table summarizing these clinical characteristics may also be useful to improve the readability of the study.
3) Discussion of obtained data is scarce. This paragraph should be profoundly revised and improved by contextualizing obtained results concerning the existing literature related to PET imaging's use in the prediction of OS before radionuclide therapy. Similarly, a pathophysiological interpretation of the obtained results might be of interest to the reader (e.g., about the observed prognostic power of SUVmin instead of SUVmean or max).
Minor points:
1) A number of methodological details are reported in the introduction of the manuscript. This paragraph should only contextualize the clinical background and the study's aim. I suggest moving these paragraphs to the methods section.
2) Some abbreviations are not spelled at their first appearance in the text (e.g., SUV).
Author Response
Major points:
1) My major concern is related to the absence of uni- and multivariable analyses in the statistical design. KM curves display the prediction of OS by a specific parameter once it is binarized. However, using the KM approach, the reader cannot understand the prediction of OS by the same parameter as a continuous variable. Similarly, without the use of multivariate analysis, the reader is not able to understand the eventual independence in the prediction of OS between the statistically significant parameters.
> Although it might not be explicitly mentioned in the manuscript, we tried to cover both uni- and multivariate methods for survival analysis. The Cox proportional hazard model (as described in page 3, section 2.3, paragraph 2) is a multivariate approach for survival analysis [1]. We used this method to identify the most significant radiomics features and to calculate the radiomics signature (RS) using the least absolute shrinkage and selection operator (LASSO) method. Consecutively, the KM estimator (as a univariate method) is used to compare the significance of the obtained RS with that of the conventional and clinical parameters. For the KM analyses, the median value of each variable throughout the cohort is used as the cut-off value to subdivide the cohort which is dynamically transformable to new patients’ data.
To wrap up, we agree that this is a major concern. Thus, we added this topic to methods (subsection 2.3., page 5) and results (subsection 3.2., pages 7, 8). Furthermore, we added figure 2 (page 8) for more clarity.
[1] Bradburn, M. J.; Clark, T. G.; Love, S. B.; Altman, D. G. Survival analysis part II: multivariate data analysis--an introduction to concepts and methods. British journal of cancer, 2003: 89(3), 431–436. https://doi.org/10.1038/sj.bjc.6601119
2) Results are scarcely detailed. In particular, a detailed description of the clinical characteristics of the study cohort is needed. A few clinical details are reported in the methods section. I suggest moving to results and implementing this section. A table summarizing these clinical characteristics may also be useful to improve the readability of the study.
> The details of the clinical characteristics are extended and moved to the results section (3.1). Table 3 is added as was recommended.
3) Discussion of obtained data is scarce. This paragraph should be profoundly revised and improved by contextualizing obtained results concerning the existing literature related to PET imaging's use in the prediction of OS before radionuclide therapy. Similarly, a pathophysiological interpretation of the obtained results might be of interest to the reader (e.g., about the observed prognostic power of SUVmin instead of SUVmean or max).
> Thank you for bringing up this issue. We carefully revised the Discussion and added references with focus in PEt for prediction of OS. We also included an idea about pathophysiological interpretation of the SUVmin.
Minor points:
1) A number of methodological details are reported in the introduction of the manuscript. This paragraph should only contextualize the clinical background and the study's aim. I suggest moving these paragraphs to the methods section.
> The methodological details are moved to the methods section as subsection 2.1. The references order has been updated consecutively.
2) Some abbreviations are not spelled at their first appearance in the text (e.g., SUV).
> Revised as requested (page 1 - Abstract)
Round 2
Reviewer 2 Report
The manuscript can be accepted in its current version.